

# Effects of vernal equinox solar eclipse on temperatures and wind directions in Switzerland

Werner Eugster[1], Carmen Emmel[1], Sebastian Wolf[2], Nina Buchmann[1], Joseph P. McFadden[3], and C. David Whiteman[4]

[1]ETH Zurich, Department of Environmental Systems Science, Institute of Agricultural Sciences, 8092 Zurich, Switzerland
[2]ETH Zurich, Department of Environmental Systems Science, Institute of Terrestrial Ecosystems, 8092 Zurich, Switzerland
[3]University of California, Santa Barbara, Department of Geography, Earth Research Institute, Santa Barbara, California, 93106–4060, USA
[4]University of Utah, Atmospheric Sciences Department, Salt Lake City, Utah, USA

*Correspondence to:* W. Eugster (eugsterw@ethz.ch)

**Abstract.** The vernal equinox total solar eclipse of 20 March 2015 produced a maximum occultation of 65.8 to 70.1% over Switzerland during the morning hours (09:22 to 11:48 CET). Skies were generally clear over the Swiss Alps due to a persistent high-pressure band between the UK and Russia associated with a rather weak pressure gradient over the continent. To assess the effects of penumbral shading on near-surface meteorology across Switzerland, air temperature data measured at 10-minute

intervals at 184 MeteoSwiss weather stations that reported air temperature at 10-minute intervals were used. Wind speed and direction data were available from 165 of these stations. Additionally, six Swiss FluxNet eddy covariance flux (ECF) sites provided turbulent measurements at 20 Hz resolution.

During maximum occultation the temperature drop was up to 5.8 K at a mountain site where cold air can pool in the topographic depression of the weather station. The bootstrapped average of the maximum temperature drops of all 184 MeteoSwiss

sites during the solar eclipse was $1.51 \pm 0.02$ K (mean $\pm$ SE). A detailed comparison with literature values since 1834 showed a temperature decrease by $2.6 \pm 1.7$ K (average of all reports) with extreme values up to 11 K. On fair weather days under weak larger scale pressure gradients, local thermo-topographic wind systems develop that are driven by small-scale pressure and temperature gradients. At one ECF site, the penumbral shading delayed the morning transition from down-valley to up-valley wind conditions, and at another site, it prevented this transition from occurring at all. Data from the 165 MeteoSwiss sites

measuring wind direction did not show a consistent pattern of wind direction response to the passing of the penumbral shadow. These results suggest that the local topographic setting had an important influence on the temperature drop and the wind flow patterns during the eclipse. Still, results tend to lend support to a recent theory that the anticyclonic cold-air outflow from the center of the eclipse only extends $\approx$1600 km outwards, with cyclonic flow beyond that distance. This contrasts with an earlier theory that the anticyclonic outflow should reach as far as $\approx$2400 km from the center of the eclipse, which would have included

all of Switzerland during the 2015 eclipse. Nevertheless, a significant cyclonic effect of the passing penumbral shadow was found in the elevation range $\approx$1700–2700 m a.s.l., but not at lower elevations of the Swiss Plateau. Thus, measurable effects of penumbral shading on the local wind system could be even found at $\approx$2000 km from the path of the eclipse (that is, Switzerland during the 2015 eclipse).



# 1 Introduction

Solar eclipses have long fascinated scientists and brought about essential scientific knowledge on the meteoriologial effects of the phenomenon. This has led to major discoveries such as the existence of helium or the high temperature of the corona (Pasachoff, 2009), but the most commonly studied effect is that on temperature at Earth's surface (Table 1, Aplin et al., 2016). Less, however, is known about the effects on local and larger scale wind directions at places where only partial occultation is observed. Here we test the hypothesis that even a partial occultation of a solar eclipse may have short-term influences on wind directions. During a solar eclipse, the new moon passes in front of the sun disk and thus reduces incoming solar radiation. This astronomic event is typically described with four phases. It begins with the first contact between the moon and the sun as seen by an observer on the Earth. With the first contact the penumbral shading begins, that is the partial shading where sunrays from one part of the sun disk are blocked by the moon whereas sunrays from the opposite side of the sun disk still reach the observer. This phase ends with the second contact when the moon completely obscures the sun and the observer is in the dark shadow of the moon. During this second phase of totality only diffuse sunlight reaches the Earth's surface. This phase ends with the maximum occultation, when the solar corona can be seen in the special case of a total eclipse. With annular and partial eclipses, the maximum occultation simply means darkest conditions. After the maximum light levels increase again until the third contact, when the transition from the umbral shadow to the less dark penumbral shadow takes place. The astronomic event ends with the fourth contact, after which meteorological conditions should no longer depend on the moon's position. The second and third contacts are only observed in the narrow band of the umbra during annular or total eclipses.

During the vernal equinox eclipse of 20 March 2015, which produced a partial occultation over the Swiss Alps, the weather conditions were excellent with mostly clear skies due to a persistent high-pressure band between the UK and Russia, with a rather weak pressure gradient over the continent (MeteoSchweiz, 2015). On normal days under weak larger scale pressure gradients, local thermo-topographic wind systems develop that are driven by small-scale pressure and temperature gradients, which are strongest in mountainous areas such as the Swiss Alps. It could hence be expected that during penumbral shading, these thermo-topographic winds would be subject to modifications superimposed by the larger scale circulation generated around the umbra of the solar eclipse. The meteorological conditions during an eclipse are expected to be the same as that of a "cyclone with a cold centre" as described by Ferrel (1890, pp. 337–342). In such a cyclone the vertical motion is reversed as compared to a typical warm centered cyclone. This leads to a (narrow) core with cyclonic rotation in the cold center of the cyclone and an anticyclonic counterflow around this core. Because of it's reversed structure, the cyclonic rotation in a cold-centered cyclone according to Ferrel (1890) is strongest in the upper troposphere and weakest near the surface, whereas the anticyclonic rotation is weakest in the upper troposphere and strongest near the Earth's surface.

Based on Ferrel's (1890) concept, Clayton (1901) empirically determined the direct influence of the occultation on wind direction within the shaded area. According to Clayton's (1901) theory, the cyclonic rotation in the center of the umbra is not detectable, but an anticyclonic outflow should be generated in the inner zone of the penumbral shadow. This zone with anticyclonic rotation is expected to extend at least 2400–3200 km (1500 to 2000 miles; Clayton, 1901) from the center of the umbra. In contrast, the outer part of the penumbra is subject to a cyclonic wind direction rotation (a further ≈1600 m; Fig. 1a).



Although Clayton (1901) concludes that this rotational pattern "confirms so well Ferrel's theory of the cold-air cyclone", he does not provide an estimate of the dimension or strength of the cold-air cyclone in the center of the umbra because no cyclonic effects could be seen in his analyses. Following Clayton (1901), Aplin and Harrison (2003) carefully assessed the penumbral winds at Reading and Camborne (UK) during the 11 August 1999 eclipse where a maximum occultation of 97% was observed. Their 1 Hz ultrasonic anemometer wind speed and wind direction measurements showed a pronounced drop in wind speed during the eclipse. The wind then changed in a cyclonic way on first contact, and later returned via an anticyclonic rotation to the synoptic wind direction after maximum occultation. Thus, their finding conformed to what is expected for the inner core around the umbra, where Clayton (1901) expected the cold-air cyclone, but did not find it in his own analysis of synoptic-scale weather maps. Aplin and Harrison (2003) thus postulated a revised model with an inner cold core of ca. 160 km around the center of the umbra containing cyclonic flow (Fig. 1b). They expected an anticyclonic rotation outside this zone that extends to ca. 1600 km (1000 mi), with a further outermost zone of cyclonic rotation then extending up to ca. 4800 km (3000 mi).

Bilham (1921) was the first to empirically confirm an indirect effect of the cooling during occultation on wind direction. He reports that "the wind showed a marked tendency to back" (as expected from cyclonic influence of the thermal wind), consistent with the cold-core cyclone in the umbral zone. There is observational evidence of thermal wind effects superimposed on near-surface winds during occultation, but no regional-scale weather prediction model has been able to reproduce this effect successfully. Prenosil (2000) simulated the 11 August 1999 eclipse in Central Europe using a hydrostatic regional weather prediction model. This model was able to produce a slight cyclonic circulation in the surface winds, but it only lasted for some minutes and thus challenges the idea that such a weak effect can be observed in field measurements.

For the Swiss Alps during the vernal equinox eclipse of 2015, there is a clear difference in expected wind direction between the Clayton (1901) and the Aplin and Harrison (2003) theories that motivated the authors to investigate whether the onset of penumbral shading leads to an anticyclonic or a cyclonic influence on near-surface wind directions. We hypothesized that during maximum occultation (66–70% across Switzerland, ≈2000 km away from the umbral center), the influence should be anticyclonic according to Clayton (1901) (Fig. 1a) or cyclonic according to Aplin and Harrison (2003) (Fig. 1b). Our aim in this paper is thus to extend the analysis of temperature drops during occultation to assess whether available wind direction data can support one of the two theories about air mass circulation inside the penumbra during the eclipse.

## 2 Material and Methods

The vernal equinox eclipse of 20 March 2015 was a Saros 120 total eclipse (partial in Switzerland), with its maximum at 09:45:39 UTC (10:45:39 CET; NASA, 2015). In Switzerland, occultation started with the first contact of the sun and moon disks at 09:21:58 CET in Geneva (western border) and ended with the fourth contact at 11:47:49 CET in Martina (eastern border). Maximum occultation ranged from 65.8% in Chiasso (southern border) to 70.1% in Bargen/Schaffhausen (northern border). The timing of maximum occultation varied from 10:29:26 to 10:35:55 CET across Switzerland (NASA, 2017). The second and third contact of a total eclipse are the entering to and exiting from the umbra, respectively. In areas with only partial occulation the second and third contact do not occur.





This eclipse has been thoroughly investigated with a focus on the mainland UK in a themed issue with 16 papers introduced by Harrison and Hanna (2016). Although all contributions are relevant, we specifically refer only to the articles with a direct link to our own study (Aplin et al., 2016; Clark, 2016; Good, 2016; Hanna et al., 2016; Burt, 2016; Pasachoff et al., 2016; Gray and Harrison, 2016; Portas et al., 2016; Barnard et al., 2016).

## 2.1 Sites and data

We used six Swiss FluxNet sites (www.swissfluxnet.ch; Table S1) with 20 Hz ultrasonic anemometer–thermometer data and 184 conventional MeteoSwiss weather stations (Table S2) across Switzerland and Liechtenstein (MeteoSwiss, 2017), of which 165 also supplemented the temperature data with wind speed and wind direction measurements (Table S3). We used data from 20 March 2015 and—where possible—the previous one or two days for reference. All three days were nearly clear, except for occasional scattered high-level clouds.

Sensors used at the MeteoSwiss station are Pt-100 thermistors for temperature measurements and Lamprecht L14512 cup anemometers with a wind vane for wind speed and wind direction measurements. Some sites alternatively use 2-D ultrasonic anemometers (MeteoSwiss, 2017). At the Swiss FluxNet sites the specific instruments included in this study are Gill HS or R3 ultrasonic anemometers (Gill Ltd., Lymington, UK), Kipp and Zonen CNR-1 four-way net-radiometers (Delft, the Netherlands) with active ventilation (Markasub, Olten, Switzerland). Only at the CH-OE2 site a Delta-T BF5 sunshine sensor (Cambridge, UK) was available for measurements of diffuse and total photosynthetic photon flux density (PPFD). PPFD is the quantum flux in the visible range 400–700 nm that plants use for photosynthesis.

For all these variables except temperature (see Section 2.2) and wind direction (see Section 2.4), the difference between 20 March and either 18 or 19 March was calculated, depending on which of the two previous days had closer to clear-sky conditions. For temperature comparisons we used two different concepts to determine the drop (see Section 2.2).

## 2.2 Calculation of temperature drop

All analyses were done with R version 3.3.1 (R Core Team, 2016). The local temperature effect at each site was estimated by fitting a local polynomial regression with a span parameter of 0.1 ("loess", a locally weighted least squares regression function in R) to each time series from 20 March, for which the measurements during the penumbral shading and the adjacent 12 minutes on both sides were excluded from the fit. The maximum difference between the measurements during penumbral shading and these model values was then determined. This approach closely follows the method used by Segal et al. (1996), or the linear approach used by Clark (2016). In a few cases (sites GRH, ROG, ULR, VAD, VSBLI; see Table S2), however, this approach failed (e.g. because of instationarities shortly before, shortly after, or during the time period of the eclipse, which can lead to erratic interpolations) and thus the simple temperature difference with respect to 19 March had to be used. We did not use this latter approach for all sites because there were substantial temperature differences between 18, 19 and 20 March 2015, despite persistent and very similar fair-weather conditions. The period of interest coincided with the peak spring snowmelt. For example, at the CH-FRU mountain grassland flux site (1000 m a.s.l.) where four phenological camera images were taken per day, the snow cover in the morning of 18 March 09:30 CET was still around 80%, but declined strongly during 18/19 March





and had less than 10% coverage by the evening of 19 March at 18:30 CET. On 20 March, the site was basically free of snow, similar to other mountainous stations that were subject to snowmelt.

## 2.3 Calculation of long-wave radiation effect

To quantify the eclipse effect on long-wave back-radiation from the sky we determined the difference between the two long-wave flux components from 20 March 2015 and the reference day before,

$$\Delta LW_x = LW_x(20 \text{ March}) - LW_x(19 \text{ March}) , \tag{1}$$

where $x$ is the incoming ($in$) or the outgoing ($out$) long-wave radiation component. Then, two linear regressions between $\Delta LW_{in}$ and $\Delta LW_{out}$ were calculated with the 1-minute averages of the CH-OE2 site, (a) for the period of the eclipse (09:26–11:42 CET), and (b) for the times of day not including the period of the eclipse.

## 2.4 Calculation of wind direction effect

The effect of penumbral shading on wind direction was determined by comparing (a) a 1-hour reference period that ended 12 minutes before the first contact with (b) the first half of penumbral shading (from first contact to maximum occultation, which was roughly 1.1 hour). For both periods, the vector-averaged mean wind direction was computed, and then the rotation angle was determined. The same procedure was repeated for the same time periods of 19 March, and the difference in rotation angle was calculated as the net effect of penumbral shading used in this study.

## 2.5 Bootstrapping of uncertainty bounds

To quantify the uncertainty of the temperature drop and wind direction effects as a function of elevation, we employed non-parametric bootstrapping with the "boot" or Bootstrap Resampling package of R in combination with the "loess" function with a span of 0.5 to describe the temperature drop or wind direction effect as a function of elevation. Elevations were binned in 10-m intervals for the bootstrap procedure, which was repeated 9,999 times. 95% statistical confidence intervals were then determined from the output.

## 2.6 Spatial interpolations

Wind direction effects were spatially interpolated with ordinary Kriging using the krige.conv function of the geoR package of R. The partial sill parameter was set to 300°, and the range parameter was set to 1000 km.

## 2.7 Image analysis

The potential effect of cloudiness during the eclipse was investigated by analyzing a sequence of phenological camera images recorded every two minutes during the eclipse (4× per day otherwise) for the brightness of the vegetation. For this, we used the ImageJ software as implemented in the Fiji image processing package, version 2.0.0 (http://imagej.net/Fiji). The vegetation





brightness is simply the normalized gray value of the image region that was manually defined as "vegetation". A brightness of 100% corresponds to a white image, and 0% is black.

## 2.8 Gamma probability distribution fit

We fit a Gamma probability density function to the histogram of the maximum cooling $\Delta T$ during the eclipse,

$$5 \quad f(\Delta T) = \frac{1}{s^a \cdot \Gamma(a)} \cdot (\Delta T - T_0)^{a-1} \cdot e^{\frac{T_0 - \Delta T}{s}} , \qquad (2)$$

where $\Delta T$ is the maximum temperature drop during an eclipse event (a positive value), and $a$ and $s$ are the shape and scale parameters of the probability density function $f(\Delta T)$, respectively. $\Gamma$ is the Gamma function, and $T_0$ is the reference value (or offset) of $\Delta T$ to fit the peak of the probability density function $f(\Delta T)$ to the data. The mean of the distribution $f(\Delta T)$ is $a \cdot s$, and its standard error is $\sqrt{a \cdot s^2}$.

## 3 Results and Discussion

### 3.1 Short-wave radiation effects

Incoming short-wave radiation can be used as a control for correct timing and magnitude of the occultation. Standard MeteoSwiss weather stations, however, only record 10-minute averages, which are too coarse for an in-depth assessment. The Swiss FluxNet sites use averaging times ranging from 10 to 30 minutes (Table S1), with the exception of the CH-OE2 crop-

15 land and CH-DAV forest sites where 1-minute averages of all four radiation components were available. Diffuse and total PPFD were also measured with the same resolution at CH-OE2, and thus we used data from CH-OE2 as an example here. The expected reduction of incoming short-wave radiation was ≈70% (Fig. 2a). The difference between measured and expected incoming short-wave radiation is –9.8 ± 2.4 W m$^{-2}$ (mean ± SD). For a second class pyranometer such as the CM3 model used in the CNR-1 radiometer, this is the order of magnitude of the accuracy. However, the timing of the radiation minimum

in Figure 2a was not exactly as expected: the theoretical radiation level during maximum occultation was reached 19 minutes before the eclipse. Most likely, this was a confounding effect due to minor high-level clouds passing at that time. The fraction of diffuse radiation started to increase shortly after the first contact on 20 March (Fig. 2b), whereas on the previous day, a curvilinear decrease was observed as expected during this early morning period with rising solar elevation. Cirrostratus clouds are the likely cause since images taken every two minutes during the occultation phase at the site do not indicate any signs of

medium and low-level clouds. This coincidence of shading by cirrostratus clouds and occultation of the sun may have led to the stronger than expected decrease in short-wave radiation levels and the earlier than expected radiation minimum.

To test this hypothesis, we made an attempt to empirically correct measured short-wave incoming radiation for potential concurrent cloud shading that leaves a trace in the fraction of diffuse radiation (Fig. 2c). The assumption we make for such a correction is that no change in the ratio of diffuse vs. direct radiation would occur due to the occultation of the sun disk alone. Thus, if we assume that the diffuse radiation (expressed as absolute radiation flux density) remains unaffected by the





cirrostratus clouds, then we can correct this effect with

$$SW_{in,\,corr} = \frac{\alpha}{\alpha_{fit}} \cdot SW_{in} \,, \qquad (3)$$

where $SW_{in}$ and $SW_{in,\,corr}$ are measured and corrected incoming short-wave radiation, and $\alpha$ and $\alpha_{fit}$ are ratios of diffuse and direct radiation for the measurements and the model, where the empirical best fit for $\alpha_{fit}$ used in the model (Fig. 2c) is

$$\alpha_{fit} = \frac{diffuse\ radiation}{direct\ radiation} = (0.674 \pm 0.002) + (0.0452 \pm 0.0002) \cdot (\Delta t)^2 \,, \qquad (4)$$

with $\Delta t$ being the time difference to local noon (12:36:45 CET on 20 March 2015) in hours. The resulting $SW_{in,\,corr}$ (red line in Fig. 2a) still does not show a symmetric effect before and after the short-wave radiation minimum. An analysis of images during that period (Fig. 2d) also indicates that the shading effect was not symmetric during occultation: the image brightness decreased very quickly after first contact, but then remained almost constant irrespective of the fraction of occultation of the sun disk. Both observations indicate a confounding effect of cirrostratus cloud passage.

## 3.2 Long-wave radiation effects

The reduction in short-wave radiation also reduces energy dissipation at Earth's surface, which in turn reduces long-wave emitted radiation ($LW_{out}$, Fig. 2e, blue line). The reduction in $LW_{out}$ is symmetric during the penumbral shadow passage, supporting our interpretation that if a cloud passage confounded the $SW_{in}$ term, then it most likely was a cloud type that affects short-wave radiation more than the long-wave radiation components. Cirrostratus may have this quality. However, the role of high clouds on the Earth's radiation budget is difficult to quantify (Boucher et al., 2013).

The reduction in $LW_{out}$ reduces the re-emitted sky radiation $LW_{in}$ (Fig. 2e, black line). The regression between $\Delta LW_{in}$ and $\Delta LW_{out}$ (Fig. 3), i.e., the differences between the respective radiation component measured on the day of the eclipse and the day before the eclipse, shows an order of magnitude difference between the penumbral shading ($\Delta LW_{in} \approx 0.24\ \Delta LW_{out}$) and the unshaded conditions ($\Delta LW_{in} \approx 2.84\ \Delta LW_{out}$).

## 3.3 Direct effects on air temperature

Although radiation effects could only be investigated at the two sites having 1-minute measurements, similar conditions were observed in 10-min data at all radiation measurement sites. All sites showed a reduction in 2-m air temperatures (Table S2). The strongest effect of –5.8 K was seen at an Alpine site (Fig. 4; VSSOR in Table S2) at 1987 m a.s.l. which was still completely snow covered during the eclipse. The topographic setting (Fig. 4) is a small basin surrounded by a larger catchment area of 7.5 km$^2$. The most important effect thus appears to be the position of the weather station. It is located in a closed topographical basin ca. 66 m below the mountain ridge. A cold-air pool building up during the eclipse could be drained towards the Rhone valley over this ridge. Had we taken the difference between 19 and 20 March for estimating the temperature effect, then this site would have yielded a difference of –8.8 K.

The mean effect over the entire elevation range covered by MeteoSwiss weather stations (Table S2) as determined by non-parametric bootstrapping is a reduction of $1.51 \pm 0.02$ K (mean $\pm$ SE; see Fig. 5). The weakest effects were found at the





lowest elevation sites (<350 m a.s.l., reduction of $0.62 \pm 0.06$ K), and the highest elevation sites, where data coverage is poor (>3150 m a.s.l., reduction of $0.69 \pm 0.03$ K).

Although the 20 March 2015 eclipse featured a partial occultation of 66–70% throughout Switzerland, the temperature effects (Fig. 5, Table S2) are comparable to temperature reductions previously reported in the literature (Fig. 6 and Table 1).

We found no clear dependence of temperature reductions on eclipse type, geographic location and other factors in the literature reports. Therefore, we summarized the data set by fitting a Gamma probability distribution to the data as shown in Figure 6. This distribution will allow researchers to quickly assess how likely it will be to find reports in the literature that exceed a given measured temperature drop during an eclipse. The parameter estimates for the probability distribution (Eq. 2) are given in Table 2. The average temperature drop reported in the literature so far was $2.6 \pm 1.7$ K (based on Eq. 2, Table 2), while the mean drop

calculated for our study was $1.5 \pm 1.0$ K. Temperature effects of this magnitude during occultation have the potential to induce thermal winds.

### 3.4 Wind direction effects

The most striking effect on wind direction was found at two Swiss FluxNet sites with high-resolution 3-D wind velocity measurements. The timing of the solar eclipse between 09:22 and 11:48 CET across Switzerland (see Section 2) coincides

with the hours when the mountain valley wind system changes wind direction by 180° under normal conditions. At the CH-DAV subalpine forest site (1639 m a.s.l.), the penumbral shading resulted in a delay of the onset of the daytime wind direction by roughly one hour (Fig. 7), whereas at the CH-AWS alpine grassland site (1978 m a.s.l.) the shading even prevented the establishment of the diurnal up-valley wind (Fig. 7). From 09:26:32 CET (first contact) to 11:46:37 CET (last contact), the short-wave radiation decreased by up to 68% (–447 W m$^{-2}$; 10-minute average) with respect to perfect clear-sky conditions

two days before (18 March 2015). This delayed the down-valley to up-valley wind direction transition that was very pronounced on both preceding days. Further, the penumbral shading hindered the onset of up-valley winds in such a way that the down-valley winds persisted even after the ending of the shading. This means that the valley wind blew in the opposite direction to what we would have predicted for comparable conditions without penumbral shading. The lack of reversal of wind direction might be an effect of the cyclonic rotation in the outer circle of the penumbra as predicted by Aplin and Harrison (2003) for

this distance of ≈2000 km from the umbra. This lack of reversal of wind direction at CH-AWS also occurred at a conventional agrometeorological weather station Michna et al. (2013) ca. 1 km further up-valley.

Although the cyclonic effect appears to be rather pronounced at CH-DAV and even more so at CH-AWS, most conventional weather stations do not show a clear signal (Figs S1–S4). In principle, only sites located on a valley bottom (Fig. S2) are expected to respond in a similar way as CH-DAV and CH-AWS. In fair weather conditions winds at slope sites (Fig. S4)

typically rotate clockwise when on the right sidewall of a valley (i.e, facing down-valley) and counterclockwise on the left sidewall, as winds undergo their diurnal transitions from along-valley to along-slope wind systems (Whiteman, 2000). Thus, the inclusion of slope sites in our analysis would confound our analysis of the wind turning associated with the shadow passage. The sites classified as "slope sites" (Table S3) turned out to be embedded in rather complex terrain for which it was impossible





to make a credible prediction which rotation should be expected. Consequently, we focus primarily on the sites not located on slopes in the following analysis.

The wind direction effect during the eclipse observed at the 165 MeteoSwiss sites included both anticyclonic and cyclonic changes (Fig. 8, green bars). If slope sites are excluded from the analysis, then a rather clear dominance of the cyclonic effect

is seen (68.7% of the remaining 112 sites; Fig. 8, black bars). In comparison to conditions during the same time of day on the day before the eclipse, the directional changes were mostly in the range –30° to –45° during the period from the first contact to maximum occultation (Fig. 8). Large direction changes exceeding –75° were less frequent than on the reference day before the eclipse.

The effect of the penumbral shading on wind direction at 10 m a.g.l. shows a strong dependence on site elevation. Figure

9 shows an elevational integration of the percentage of sites showing cyclonic influence during the penumbral shading as expected according to Aplin and Harrison (2003) (Fig. 1b). There is only one elevation zone (1708–2730 m a.s.l., 14 sites) in which a significant cyclonic rather than anticyclonic influence (p<0.05 according to bootstrapped uncertainty bounds) was found. The eclipse effect seen at the sites in this elevation zone is clearly in support of the revised theory by Aplin and Harrison (2003), which reduced the extent of the anticyclonic outflow around the umbra from ≈2400 km Clayton (1901) to ≈1600 km

(Fig. 1). Interestingly, however, Hanna et al. (2016) did not find any discernible effect of this eclipse on wind directions in the UK and Iceland, and thus deduce that there was no evidence of an eclipse cyclone. In contrast to Switzerland, the UK observed a fair share of cloudiness (see satellite imagery in Hanna et al., 2016) which may have muted some meteorological responses to the occultation, as Burt (2016) noted. But if the analysis is constrained to sites with clear-sky conditions, Gray and Harrison (2016) found a clear cyclonic effect of approximately 20–40° in the comparison of surface measurements with

forecast model simulations which were ignorant of the eclipse. Which such a cyclonic effect is what would be expected for Switzerland following the theory by Aplin and Harrison (2003), it contradicts that theory for the geographic location of the British Islands, which should observe anticyclonic modification of the wind direction under both Clayton (1901) and Aplin and Harrison (2003) theories. Hence, Gray and Harrison (2016) offer a new interpretation similar to the nocturnal low level jet. This interpretation is however not likely to explain conditions in the complex terrain fo the Swiss Alps, and hence we did not

further adopt this interpretation here.

### 3.5 Spatial patterns of wind direction effects

To test if there exists a geographically consistent pattern of wind direction changes across Switzerland and Liechtenstein we performed a spatial interpolation (Fig. 10). For the reasons given above, we again focused on sites that are not located on slopes (Fig. 10b). The resulting map, however, is relatively similar to the one using all sites (Fig. 10a).

If all stations are considered irrespective of their individual topographic environment (Fig. 10a), then a complex pattern emerges that does not show a clear spatial structure that could be related to the passing of the penumbral shadow. Also valley bottom sites (Fig. 10e) show a mixture of anticyclonic effects in the west, south of the Alps, and eastern Switzerland, with cyclonic effects seen between these three anticyclonic areas. Sites on flat ground without clear topographic influences (Fig. 10c) show cyclonic effects in the center of Switzerland, surrounded by anticyclonic effects namely in the Valais (southwestern





Switzerland) and the northeast. Mountain top and hilltop sites (Fig. 10d) show yet another pattern, with anticyclonic influences seen mostly in the western half of Switzerland and cyclonic effects in the eastern half. The only group of sites showing a consistent spatial pattern are those in the elevation range 1708–2730 m a.s.l. (Fig. 10f), the ranges with a statistically significant preference for cyclonic effects in Figure 9. This cyclonic effect is seen across most of Switzerland, with the exception of a few

high mountain sites in the Grisons (eastern Switzerland), the part of the study domain that is farthest away from the trajectory of the eclipse (Fig. 1). This elevation range corresponds to the elevation range where the influence of the topographical roughness of the Swiss mountains vanishes in vertical radiosonde profiles (at ≈2500 m, Wanner and Furger, 1990).

Taken together, these findings lend support to the hypothesis that Switzerland was in the cyclonic part of the penumbral shading as expected from the Aplin and Harrison (2003) theory (Fig. 1b), not within the anticyclonic part as would be expected

from the Clayton (1901) theory (Fig. 1a). Although no theory exists on how the transition zone between the inner anticyclonic and the outer cyclonic ring around the umbra should affect local wind directions, our analysis indicates that the effect is most likely a combination of distance from the center combined with the meso- and micro-scale topography around the site. This is not unexpected if the net effect of the shading is weak. The fact that there is a significant preference for cyclonic effects at sites in the elevation range 1708–2730 m a.s.l., whereas there is no significant difference between anticyclonic and cyclonic

effects at other elevations, suggests that the reduced dimensions of the anticyclonic band around the umbra as suggested by Aplin and Harrison (2003) is more likely to be correct than the original Clayton (1901) model, in which Switzerland should have experienced a shift from cyclonic to anticyclonic influence as the occultation progressed towards its local maximum.

### 3.6   Comparison with findings from other eclipses

The present study expands upon previously published analyses of the eclipse effect using multiple stations in a given region.

For example Brazel et al. (1993) performed a similar analysis of temperature effects using 16 weather stations in the Phoenix, Arizona, metropolitan area, but were unable to analyse wind direction effects. They mention two reasons why this was not feasible: (1) the eclipse occured in the morning when the wind flow tends to reverse due to topographic heating in the Salt River Valley, and (2) several of the available stations did not record wind direction, but only wind speed. The first point (given with reference to Frenzel, 1963) is typical for any topographically varying region on the globe and exactly matches the

conditions experienced in this study. This effect was also emphasized by Anderson (1999), Vogel et al. (2001), and Sjöblom (2010): the surface cooling can trigger downslope and katabatic winds in mountainous regions, such as the Alps, the Arctic islands and Antarctica.

On 18 to 20 March 2015, synoptic pressure conditions over the European Alps showed a persistent high-pressure band between the UK and Russia, providing an excellent basis for comparing conditions during the solar eclipse with previous days.

Although such comparisons are one of the most used approaches to quantify the effects of solar eclipses (Owen and Jones, 1927; Shur, 1984; Brazel et al., 1993; Jain et al., 1997; Dutta et al., 1999; Dolas et al., 2002; Gorchakov et al., 2007; Chernogor, 2008; Gorchakov et al., 2008; Sjöblom, 2010; Bala Subrahamanyam and Anurose, 2011; Bala Subrahamanyam et al., 2011; Muraleedharan et al., 2011; Bala Subrahamanyam et al., 2012; Murthy et al., 2013), in most cases weather conditions are rather variable and nonideal for direct comparisons. In Switzerland, the 20 March 2015 eclipse occurred when snow cover





was disappearing at montane locations around 1000 m a.s.l. and thus air temperatures varied more strongly from day to day than what would be desirable for estimating temperature effects. Therefore, we employed the less widely used approach by Segal et al. (1996) to fit a curvilinear interpolation over the period of occultation from first to fourth contact. This approach is typical for assessing effects on atmospheric constituents such as ozone (Tzanis et al., 2008). A third approach to deal with the

variability of measurements during reference days is to take the 30-day hourly median values for comparison (Gerasopoulos et al., 2008). All three approaches have advantages and disadvantages. In our case, the direct comparison with the day before the eclipse during which similar synoptic weather conditions persisted would have led to a more pronounced temperature drop by an additional 1.3 K on average. The relationship between the temperature drop of all sites reported in Table S2 as used in our study and the absolute difference to the day before the eclipse was

$\Delta T_{19} = -2.06 \pm 0.17 + (0.48 \pm 0.09)\Delta T$ ,                                                               (5)

with $\Delta T$ being the temperature drop used in this study and $\Delta T_{19}$ being the alternative calculation of $\Delta T$ as the difference between the measurements made on 20 March and the same time of day on 19 March (the day before the eclipse; $p < 0.001$, adj. $R^2 = 0.124$).

The choice of method to determine the temperature drop—and any other variable of interest affected by the penumbral

shading—has a substantial effect on the result, not only in our study, but also in other published results (e.g. Table 1). Only 9% of the studies considered have reported stronger temperature effects than the most extreme 5.8 K drop at the Sorniot–Lac Inférier site reported here, although most temperature drop reports in the literature relate to full occultation during total and annular eclipses. With future eclipses it may now become possible to engage citizen scientists to determine the temperature drop during an eclipse and relate it to the probability distribution presented in Table 3, which would allow them to put their

measurements into context (see also Portas et al., 2016; Barnard et al., 2016).

Differences in temperature drops among different eclipse events are expected to depend on solar elevation at the time of the eclipse (Reynolds, 1937). Here we showed that although the eclipse happened in the morning hours and although occultation was only partial at all sites in Switzerland, substantial drops in air temperatures at several Swiss sites (Table S2) exceeded those observed at other locations with higher solar angles (Table 1). Thus, our results suggest that the topographic setting, not the

geographic position on the globe, may actually be the most important determinant of the temperature drop for individual sites.

With respect to wind direction effects only few studies are available for comparison, with Clayton (1901) still being a classical reference after more than one century. Regional weather forecast models are now run at a spatial resolution that should theoretically allow simulations of the effect of penumbral shading on wind speed (not assessed here due to inconsistent variability among the sites) and wind direction (as presented in Fig. 10). Gray and Harrison (2012) followed this approach for

the southern UK during the 11 August 1999 eclipse but found that the umbral shading did not produce clear effects on wind vectors. They concluded that the primary response of the model was restricted to temperatures. Similarly, Prenosil's (2000) simulations for Central Europe showed model responses for temperature and humidity, but not for near-surface winds. Few wind observations are available for testing of such models, and Prenosil (2000) concluded that special observation campaigns with very accurate sensors would be required to make progress. The six Swiss FluxNet sites provide detailed measurements, and





many more sites are available globally via Fluxnet (Baldocchi et al., 2001). However, the Fluxnet sites provide only aggregated 30-minute averages, not the raw data at original resolution. Of our six sites with available high-frequency measurements, two sites (CH-DAV and CH-AWS, Fig. 7) showed a clear wind direction effect during the penumbral shading, even though the distance to the center of the eclipse was on the order of 2000 km.

## 4   Conclusions

Temperature responses to the eclipse were quite strong (up to 5.8 K cooling, $1.51 \pm 0.02$ K on average), especially in basin topography where the cold air from the occultation can pool. Our results suggest that the topographic setting is the most important determinant of the temperature drop for individual sites. Effects on wind directions were variable and site-specific as well. Thermo-topographic winds generated in the penumbral zone of a solar eclipse can modify local wind systems, either by delaying of the onset of the diurnal wind shift (as seen at the CH-DAV site) or by preventing the establishment of an up-valley wind in mountainous areas (at the CH-AWS site). These effects can occur at a considerable distance from the umbral path and two theories exist on the wind direction effect at a distance of $\approx 2000$ km from the center of the eclipse (Fig. 1). According to Clayton (1901), an anticyclonic outflow should extend up to 2400 km from the center of the eclipse, which would include all of Switzerland during the 20 March 2015 eclipse. The modified theory by Aplin and Harrison (2003) reduces this extent to 1600 km, which would exclude all of Switzerland during the 20 March 2015 eclipse and leave it in the outermost ring with cyclonic counterflow. Our analysis indicates a significant preference for the latter, reduced dimensions, at least for the elevation range 1706–2750 m a.s.l. (the elevations at which the typical influence of the roughness of the mountain topography tends to vanish). In this elevation zone the influence of the penumbral shading on near-surface wind directions is detectable even at distances greater than 2000 km from the center of the umbra, at places where maximum occultation during an eclipse is as low as 66–70%.

## 5   Data availability

Our data policy is available under http://www.gl.ethz.ch/research/data-archive.html. Aggregated Swiss FluxNet data are directly available from the European Flux Database Cluster (http://gaia.agraria.unitus.it/pi-area; 30-min averages). Binary raw data files, ancillary data at finer resolution than 30-min, and phenological camera images are available from the corresponding author on request. The MeteoSwiss data are directly available via their IDAWEB interface (https://gate.meteoswiss.ch/idaweb?language=en; registration required; freely available for non-commercial research projects).

*Author contributions.* WE designed the study, carried out all analyses, wrote and revised the manuscript. CDW provided essential topo-climatological input and advice for the analyses. All co-authors were involved in writing and contributed to the article with feedback and critique.



*Competing interests.* None.

*Acknowledgements.* This study made extensive use of standard meteorological data collected by MeteoSwiss, the Swiss Federal Office for Meteorology and Climatology, which also contains sites from the Swiss Airforce (Kdo LVb FULW 34) and the Swiss Federal Research Institute WSL. We acknowledge the use of these data via the IDAWEB online database. We thank Jon Eugster, University of Zurich, for

5   mathematical support with probability distribution model implementation. The Swiss National Science Foundation (SNF) supported research at the CH-OE2 Swiss FluxNet site via grant 146373, at the CH-DAV site via grant 148992 (ICOS), and at the CH-CHA and CH-FRU sites via grant 154245.




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





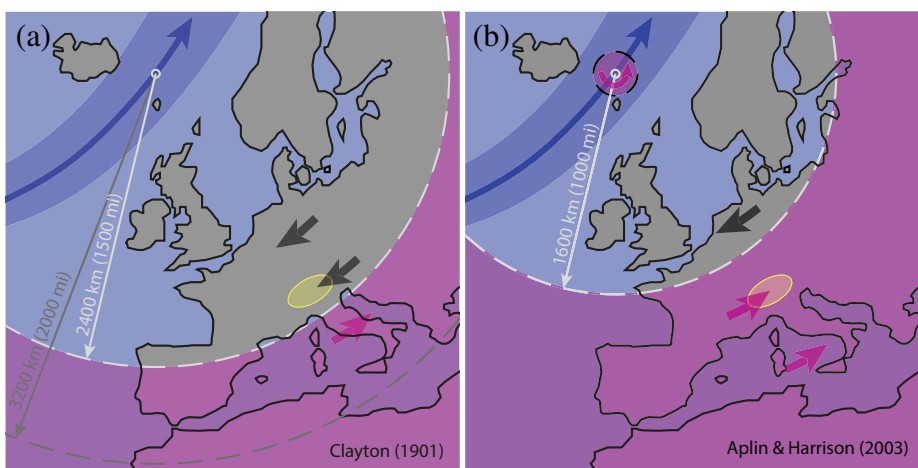

**Figure 1.** Theoretical effects of penumbral shading on wind direction. Wind directions around the center of the umbra (east of Iceland) according to the theory of Clayton (1901) (left) and Aplin and Harrison (2003) (right), schematically drawn for the time of maximum occultation over the Swiss Alps (yellow area). The eclipse trajectory is shown with a blue arrow, and the umbral path (100% occultation) is shown with a blueish band. The schematic shows the time of greatest eclipse with it's center indicated by a gray circle.





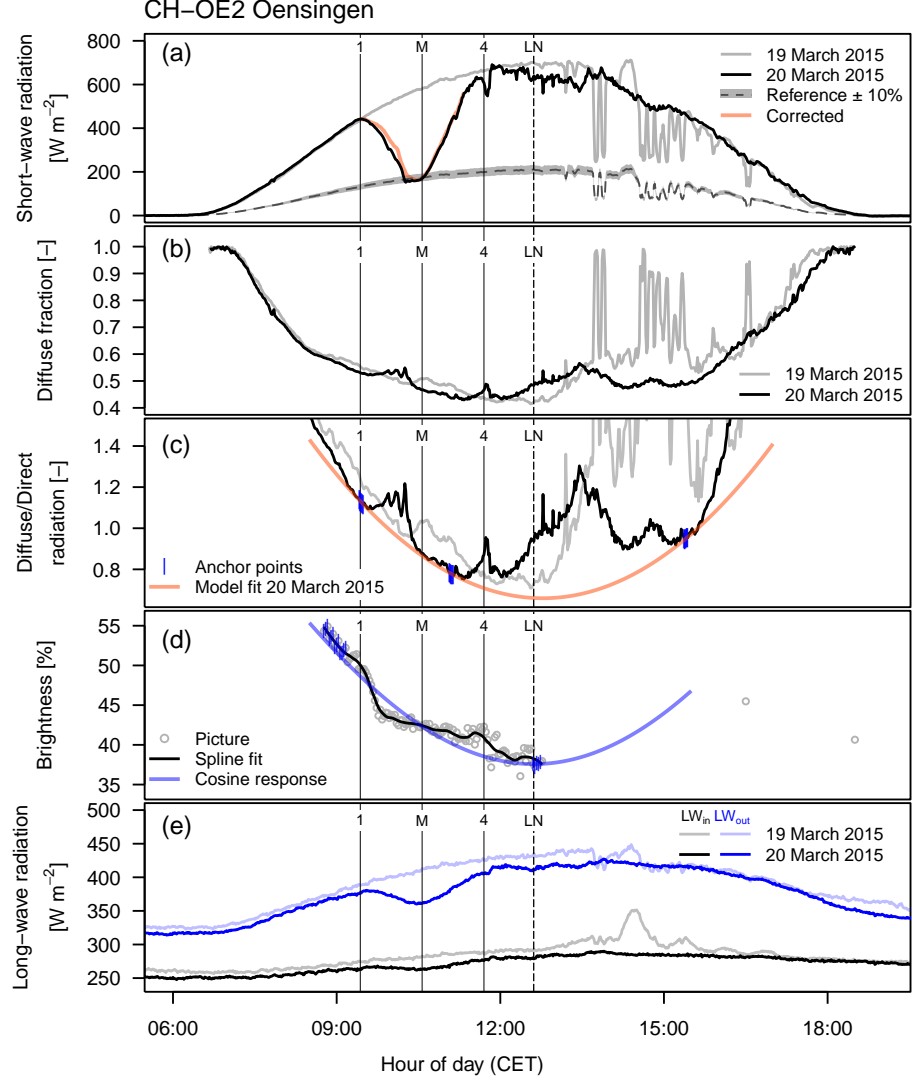

**Figure 2.** Radiation effects at the CH-OE2 Swiss FluxNet site. (a) Incoming short-wave radiation, (b) fraction of diffuse radiation, (c) ratio between diffuse and direct radiation, (d) evolution of brightness of the vegetation during the eclipse, and (e) long-wave radiation components. The four vertical lines indicate first contact (1), maximum occultation (M), last contact (4), and local noon (LN). The dashed reference curve in (a) is the measurement from 19 March 2015 multiplied with 0.6996, the theoretical maximum occultation at the site. The red model curve in (c) was fitted to the three periods with blue datapoints to correct (a) for cloud passage during the eclipse. The blue curve in (d) is a corresponding fit to the image data. $LW_{in}$ and $LW_{out}$ in (e) are incoming and outging long-wave radiation, respectively. See text for details.



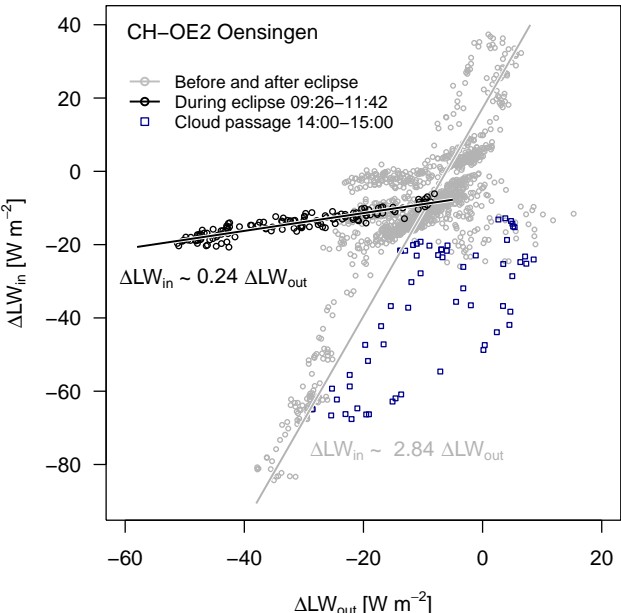

**Figure 3.** Long-wave back radiation effect during the eclipse (black symbols and regression line) in comparison to conditions before and after the eclipse (gray symbols and regression line). The differences $\Delta LW$ between 1-minute averages from the day before the eclipse and the same time during the day of the eclipse are shown ($\Delta LW_{in}$ in relation to $\Delta LW_{out}$; see Section 2.3, Eq. 1). Measurements made during a period with cloud passage on 19 March 2015 (blue symbols) were excluded from analysis.




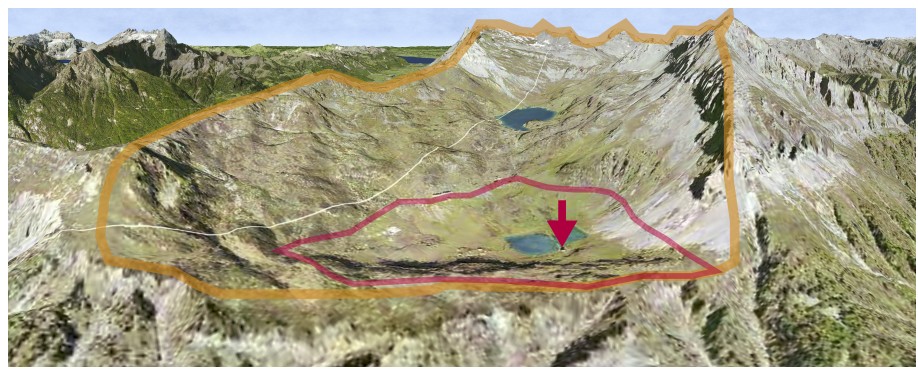

**Figure 4.** Oblique view of the surroundings of the Sorniot–Lac Inférieur de Fully weather station. The red arrow marks the location of the station, the red line marks the drainage area up to the lowest pass, and the orange line marks the full drainage area. The area was completely covered with snow and the lake was frozen during the eclipse. Imagery from Atlas of Switzerland V3 (Hurni, 2010), © 2016 swisstopo (JD100042).




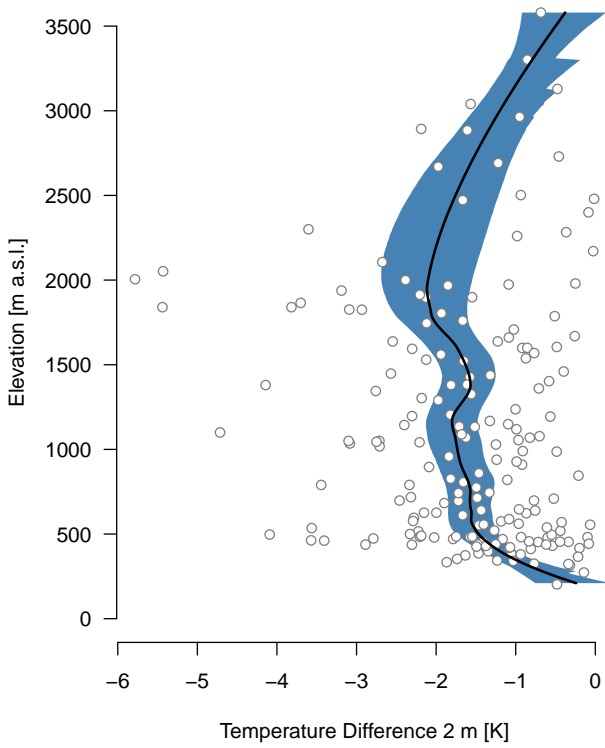

**Figure 5.** Temperature reduction maximum during the solar eclipse at 184 weather stations in Switzerland and Liechtenstein (open circles) that record 10-minute averages. The elevation profile (bold line) and its 95% confidence interval (blue band) were estimated using nonparametric bootstrapping. See Section 2.2 and Table S2 for details.




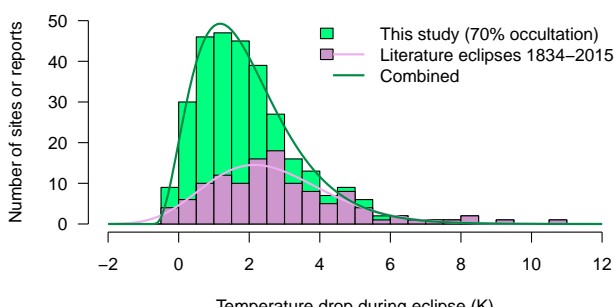

**Figure 6.** Histogram of temperature reduction at all sites included in this study and those reported in the literature. The stacked bars show number of sites of this study (green bars) on top of those for literature reports (violet bars). The solid lines show the best fit of the scaled probability distributions (Eq. 2, Table 2) of values reported in the literature (violet line) and the combination of literature data with values reported in this study (dark green line).

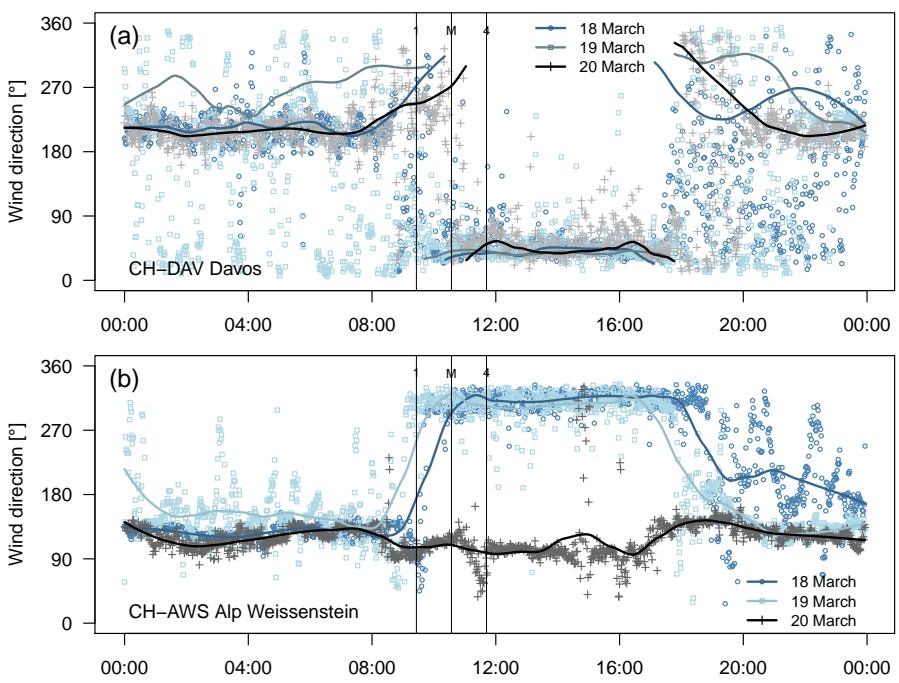

**Figure 7.** Wind direction as a function of time (a) at CH-DAV and (b) at CH-AWS on 18, 19 and 20 March, showing a delayed wind direction reversal at CH-DAV on 20 March (a), and a suppression of the typical wind reversal at CH-AWS on 20 March (b). From 1-minute average data. The bold lines are local polynomial regression (loess) fits.





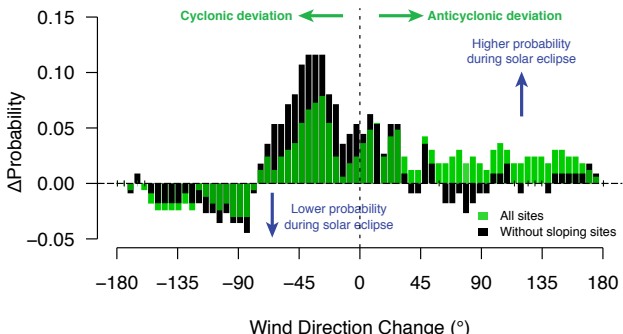

**Figure 8.** Deviation of wind direction changes (range –180° to 180°) during the eclipse expressed as ΔProbability with respect to a random uniform distribution. The wind direction change is the difference between the first half of the eclipse in comparison with the corresponding time period on 19 March.





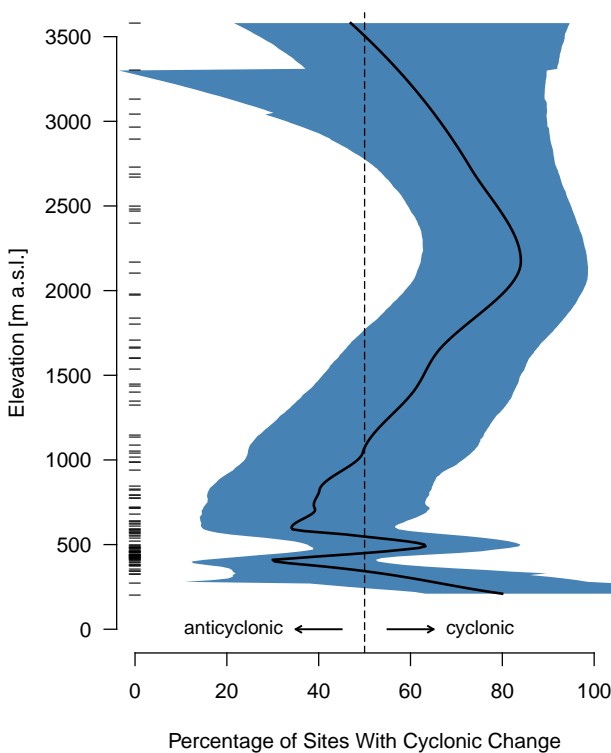

**Figure 9.** Elevation dependence of cyclonic and anticyclonic influences during penumbral shading. Using MeteoSwiss sites the percentage of sites showing cyclonic or anticyclonic effects was determined and elevational best estimates (bold line) and uncertainty of the estimate (90% confidence interval in blue) were estimated using nonparametric bootstrapping. The vertical dashed line at 50% indicates the insignificant random outcome. Each horizontal mark near the elevation axis represents one weather station.





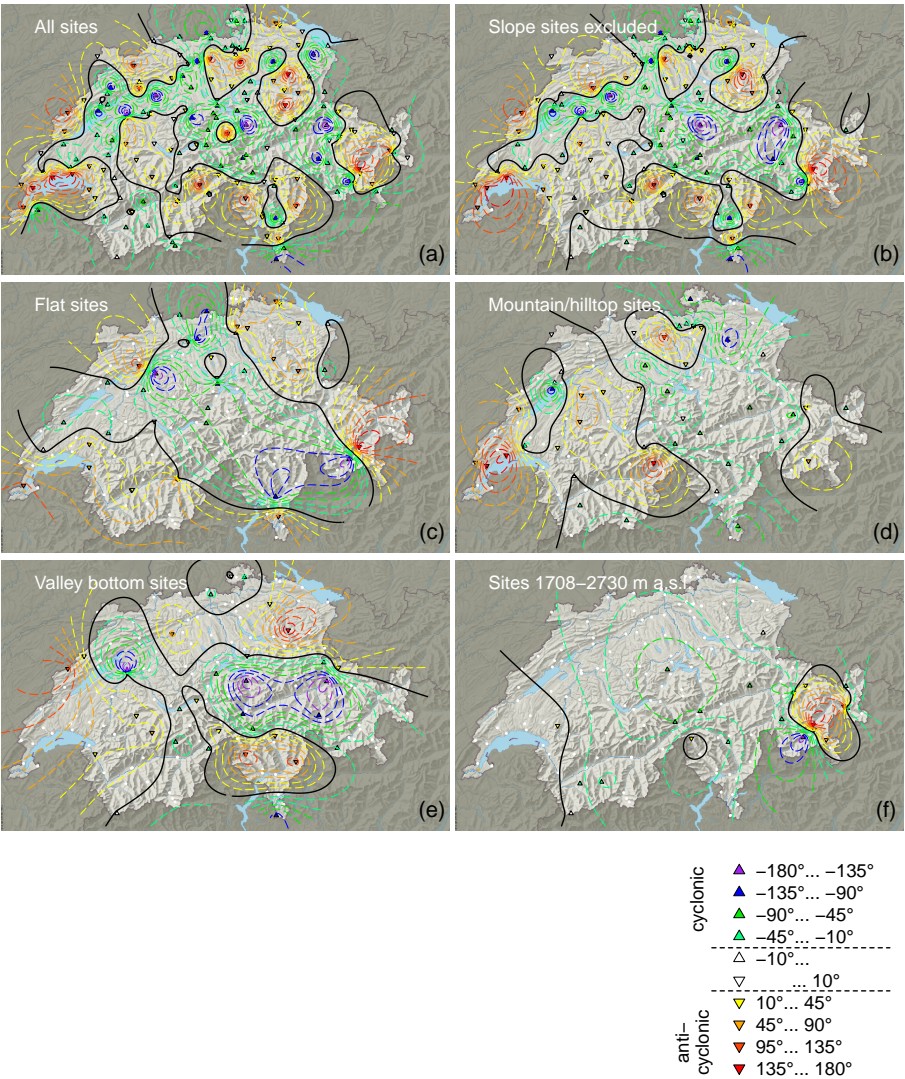

**Figure 10.** Spatial distribution of wind direction changes over Switzerland during the 20 March 2015 eclipse. Yellowish to reddish colors indicate anticyclonic rotation, while greenish to bluish colors indicate cyclonic rotation. The bold and broken isolines show lines of equal rotation angle at 20° intervals. The bold line separates areas with cyclonic from anticyclonic wind direction changes. Base map from Atlas of Switzerland V3 (Hurni, 2010), © 2016 swisstopo (JD100042).



**Table 1.** Compilation of literature reports on temperature drops during maximum occultation of the eclipse since 1834. Most reports are from total or annular eclipses, but a few studies also report values from partial eclipses or partial occultation at a given locality. More observations are tabulated in Aplin et al. (2016), but not all these reports allow the calculation of the temperature drop.

| Date | Location \| Additional Information | Temperature Drop | Reference |
|---|---|---|---|
| 1834-11-30 | Boston, Mass., USA | 0.3 K | Anonymous (1834) |
| 1896-08-09 | Vadsö, Norway (70°04′ N) | 1.0–1.6 K | Mill (1896) |
| 1896-08-09 | Vadsö, Norway (70°04′ N) | 3.1 K | Ward (1896) |
| 1905-08-30 | Burgos | 8.3 K | Reynolds (1937) |
| 1918-06-08 | Goldendale, Washington, USA | 3.6 K | Anderson (1999) |
| 1921-04-08 | Bexley Heath | 1.1 K | Bilham (1921) |
| 1921-04-08 | Bristol | 4.2 K | Bilham (1921) |
| 1921-04-08 | Nottingham (Lenton Fields) | 3.0 K | Bilham (1921) |
| 1921-04-08 | Prestou (Hoghton) | 1.7 K | Bilham (1921) |
| 1927-06-29 | Bangor, UK | 0.5 K | Owen and Jones (1927) |
| 1927-06-29 | English eclipse, cloudy | nothing remarkable | Reynolds (1937) |
| 1927-06-29 | Southport | 0.5 K | Ashworth (1927) |
| 1932-08-31 | Canadian eclipse, cloudy | very small fall | Reynolds (1937) |
| 1936-06-19 | Chios | 1 K | Reynolds (1937) |
| 1936-06-19 | Omsk | 5 K | Reynolds (1937) |
| 1936-06-19 | on steam ship Strathaird | 1.5 K | Reynolds (1937) |
| 1936-06-19 | Portugal | 2.7–3.3 K | Reynolds (1937) |
| 1936-12-13 | New Plymouth, New Zealand | 4.2 K | Hayes (1937) |
| 1970-03-07 | Lee, Florida, USA | 3.2 K | Anderson et al. (1972) |
| 1973-06-30 | Chinguetti, Mauritania | 3.5 K | Anderson and Keefer (1975) |
| 1973-06-30 | Chinguetti, Mauritania | 2.5 K | Anderson and Keefer (1975) |
| 1973-06-30 | Chinguetti, Mauritania | 2.5 K | Anderson and Keefer (1975) |
| 1979-02-26 | Hecla, Manitoba, Canada | 2.0 K | Anderson (1999) |
| 1984-05-30 | 'in Georgia' | 7.8 K | Menke (1988) |
| 1988-03-18 | Ship, coast of Karimata island | 2.2 K | Menke (1988) |
| 1991-07-11 | Agriculture/golf, wet fraction 1.00, albedo 0.20–0.25 | 1.40 K | Brazel et al. (1993) |
| 1991-07-11 | Costa Rica | no info | Fernández et al. (1993b) |
| 1991-07-11 | Costa Rica, Damas | 4.7 K | Fernández et al. (1993a) |
| 1991-07-11 | Costa Rica, Fabio Baudrit Experimental Station | 5.5 K | Fernández et al. (1993a) |
| 1991-07-11 | Costa Rica, Liberia, Alajuela and Palmar Sur | 3.0–3.5 K | Fernández et al. (1993a) |
| 1991-07-11 | Costa Rica, Limón | 3.0 K | Fernández et al. (1993a) |
| 1991-07-11 | Costa Rica, Puntarenas | 2.7 K | Fernández et al. (1993a) |
| 1991-07-11 | Costa Rica, Santa Cruz and Filadelfia | 2.0–2.5 K | Fernández et al. (1993a) |
| 1991-07-11 | Costa Rica, Tárcoles | 8.5 K | Fernández et al. (1993a) |



| Date | Where \| Additional Information | Temperature Drop | Reference |
|---|---|---|---|
| 1991-07-11 | Desert, wet fraction 0.00, albedo 0.27 | 2.65 K | Brazel et al. (1993) |
| 1991-07-11 | Fresno, California, USA, cotton field | ca. 4.5 K | Mauder et al. (2007) |
| 1991-07-11 | Industrial/airport, wet fraction 0.07, albedo 0.1 | 1.38 K | Brazel et al. (1993) |
| 1991-07-11 | Residential/commercial, wet fraction 0.47, albedo 0.20–0.25 | 1.93 K | Brazel et al. (1993) |
| 1994-05-10 | Ames, IA, USA | 2.3 K | Segal et al. (1996) |
| 1994-05-10 | Boulder, CO, USA | 2.2 K | Segal et al. (1996) |
| 1994-05-10 | Chicago, IL, USA | 6.1 K | Segal et al. (1996) |
| 1994-05-10 | Estes Park, CO, USA | 3.6 K | Segal et al. (1996) |
| 1994-05-10 | Ft. Collins, CO, USA | 2.2 K | Segal et al. (1996) |
| 1994-05-10 | Keenesburg, CO, USA | 3.0 K | Segal et al. (1996) |
| 1994-05-10 | Lakewood, CO, USA | 2.7 K | Segal et al. (1996) |
| 1994-05-10 | Lamberton, MN, USA | 3.1 K | Segal et al. (1996) |
| 1994-05-10 | Longmont, CO, USA | 2.8 K | Segal et al. (1996) |
| 1994-05-10 | Loveland, CO, USA | 3.3 K | Segal et al. (1996) |
| 1994-05-10 | Morris, MN, USA | 2.3 K | Segal et al. (1996) |
| 1994-05-10 | Norman, OK, USA | 3.6 K | Segal et al. (1996) |
| 1994-05-10 | Nowata, Oklahoma, USA | 3.0 K | Anderson (1999) |
| 1994-05-10 | Nunn, CO, USA | 1.9 K | Segal et al. (1996) |
| 1994-05-10 | Rollinsville, CO, USA | 2.3 K | Segal et al. (1996) |
| 1994-05-10 | Sedalia, MO, USA | 4.2 K | Segal et al. (1996) |
| 1994-05-10 | Springfield, IL, USA | 6.1 K | Segal et al. (1996) |
| 1994-05-10 | St. Paul, MN, USA | 1.5 K | Segal et al. (1996) |
| 1994-05-10 | Waseca, MN, USA | 3.7 K | Segal et al. (1996) |
| 1994-05-10 | White Sands, New Mexico | 5.5 K | Anderson (1999) |
| 1994-05-10 | White Sands, New Mexico | 0.4 K | Anderson (1999) |
| 1994-11-03 | Coronel Oviedo, Paraguay | 3.3 K | Fernández et al. (1996) |
| 1995-10-24 | Neem ka Thana, India | 3 K | Jain et al. (1997) |
| 1995-10-24 | New Delhi, India | 1.5 K | Jain et al. (1997) |
| 1995-10-24 | New Delhi, India | 6–8 K | Jain et al. (1997) |
| 1995-10-25 | Hyderabad, India | 9–10 K | Dutta et al. (1999) |
| 1998-02-26 | Sinamaica, Venezuela | 5 K | Nufer and Gfeller (1998) |
| 1999-08-11 | Akola, Central India | 1–2 K | Dolas et al. (2002) |
| 1999-08-11 | Kharkiv, Ukraine, max. occultation 0.73 | 7.3 K | Chernogor (2008) |
| 1999-08-11 | Modeling study, Central Europe | average 3.5 K | Gross and Hense (1999) |
| 1999-08-11 | Modeling study, Central Europe | peak up to 5 K | Gross and Hense (1999) |
| 1999-08-11 | Silsoe, Bedfordshire, UK, soil temperature at 10 mm depth | 1.6 K | Leeds-Harrison et al. (2000) |
| 1999-08-11 | Silsoe, Bedfordshire, UK, under grass temperature | 0.5 K | Leeds-Harrison et al. (2000) |





| Date | Where | Additional Information | Temperature Drop | Reference |
|------|-------------------------------|------------------|---------------------------------------|
| 1999-08-11 | Southern UK | up to 3 K | Gray and Harrison (2012) |
| 1999-08-11 | Szczawnica, Poland | 11 K | Szalowski (2002) |
| 2001-06-21 | Lusaka, Zambia | 5.38 ± 0.04 K | Penaloza-Murillo and Pasachoff (2015) |
| 2003-05-31 | Kharkiv, Ukraine, max. occultation 0.64 | 2.1 K | Chernogor (2008) |
| 2005-10-03 | Kharkiv, Ukraine, max. occultation 0.24 | 1.3 K | Chernogor (2008) |
| 2006-03-29 | central Greece | 2.7 K | Nymphas et al. (2009) |
| 2006-03-29 | Finokalia, Greece | 2.3 K | Founda et al. (2007) |
| 2006-03-29 | Ibadan, Nigeria, 1 m | 1.6 K | Nymphas et al. (2009) |
| 2006-03-29 | Ibadan, Nigeria, 12 m | 0.8 K | Nymphas et al. (2009) |
| 2006-03-29 | Ibadan, Nigeria, 6 m | 1 K | Nymphas et al. (2009) |
| 2006-03-29 | Kastelorizo, Greece | 2.3 K | Founda et al. (2007) |
| 2006-03-29 | Kharkiv, Ukraine, max. occultation 0.77 | 2.3 K | Chernogor (2008) |
| 2006-03-29 | Kislovodsk, Russia | 3 K | Gorchakov et al. (2008) |
| 2006-03-29 | Kislovodsk, Russia, 600 m a.s.l. | 2 K | Gorchakov et al. (2007) |
| 2006-03-29 | Kislovodsk, Russia, surface atmospheric layer | 3.4 ± 0.5 K | Gorchakov et al. (2007) |
| 2006-03-29 | Manavgat, Turkey | 5 K | Stoev et al. (2008) |
| 2006-03-29 | Markopoulo (Athens), Greece | 2.7 K | Founda et al. (2007) |
| 2006-03-29 | northern Greece | 3.9 K | Nymphas et al. (2009) |
| 2006-03-29 | Palini (Athens), Greece | 1.6 K | Founda et al. (2007) |
| 2006-03-29 | Penteli (Athens), Greece | 2.7 K | Founda et al. (2007) |
| 2006-03-29 | Side, Turkey | 5 K | Pleijel (2009) |
| 2006-03-29 | southern Greece | 2.3 K | Nymphas et al. (2009) |
| 2006-03-29 | Thessaloniki, Greece | 3.9 K | Founda et al. (2007) |
| 2006-03-29 | Thission (Athens), Greece | 2.6 K | Founda et al. (2007) |
| 2006-03-29 | Athens, Greece | 0.7 K | Tzanis et al. (2008) |
| 2006-03-29 | Ibadan, Nigeria | 2.2 K | Economou et al. (2008) |
| 2008-08-01 | Svalbard, Norway | 0.3–1.5 K | Sjöblom (2010) |
| 2010-01-15 | Gadanki, India, –0.10 m | 3.0 K | Venkat Ratnam et al. (2010) |
| 2010-01-15 | Gadanki, India, –0.20 m | 1.3 K | Venkat Ratnam et al. (2010) |
| 2010-01-15 | Gadanki, India, 0.00 m | 5.4 K | Venkat Ratnam et al. (2010) |
| 2010-01-15 | Gadanki, India, 0.05 m | 5.0 K | Venkat Ratnam et al. (2010) |
| 2010-01-15 | Gadanki, India, 12 m | 2.5 K | Venkat Ratnam et al. (2010) |
| 2010-01-15 | Gadanki, India, 4 m | 5 K | Venkat Ratnam et al. (2010) |
| 2010-01-15 | Gadanki, India, 8 m | 3 K | Venkat Ratnam et al. (2010) |
| 2010-01-15 | Gadanki, India, surface | 5.8 K | Venkat Ratnam et al. (2010) |



| Date | Where | Additional Information | Temperature Drop | Reference |
|---|---|---|---|
| 2010-01-15 | Kanyakumari, India | 4 K | Murthy et al. (2013) |
| 2010-01-15 | Ramanathapuram, India | 1 K | Murthy et al. (2013) |
| 2010-01-15 | Thiruvananthapuram, India | 1.2 K | Bala Subrahamanyam and Anurose (2011) |
| 2010-01-15 | Thiruvananthapuram, India, over cassava | 4 K | Murthy et al. (2013) |
| 2010-01-15 | Thrissur, India | 2 K | Murthy et al. (2013) |
| 2010-01-15 | Thumba, India | 2 K | Murthy et al. (2013) |
| 2010-01-15 | Thumba, India | 1.2 K | Bala Subrahamanyam et al. (2012) |
| 2010-01-15 | Tirunelveli, India | 3.2 K | Murthy et al. (2013) |
| 2015-03-20 | Mainland UK, maximum drop (of 266 sites) | 4.23 K | Clark (2016) |
| 2015-03-20 | Mainland UK, median drop (of 266 sites) | 1.02 K | Clark (2016) |
| 2015-03-20 | Mainland UK, minimum drop (of 266 sites) | 0.03 K | Clark (2016) |
| 2015-03-20 | Mainland UK, mean drop (of 76 sites) | $0.83 \pm 0.63$ K | Hanna et al. (2016) |
| 2015-03-20 | Mainland UK, mean drop, clear sky (14 sites) | $0.91 \pm 0.78$ K | Hanna et al. (2016) |
| 2015-03-20 | Mainland UK, mean drop, cloudy sky (16 sites) | $0.31 \pm 0.40$ K | Hanna et al. (2016) |
| 2015-03-20 | Svalbard, Norway | 0.3–1.5 K | Pasachoff et al. (2016) |
| 2015-03-20 | Switzerland, 184 stations | 1.5 K | *This study* |
| 2015-03-20 | Sorniot–Lac Inférier (Switzerland, most extreme drop) | 5.8 K | *This study* |





**Table 2.** Fitting parameters of gamma distribution (Eq. 2) fitted to empirical histograms of temperature drops $\Delta T$ during the eclipses and mean $\Delta T$. All values are best estimates $\pm$ standard error of the estimate. Values in italics indicate that the parameter estimates were not significantly different from zero (p > 0.05).

|  | Offset | Shape | Scale | mean $\Delta T$ |
|---|---|---|---|---|
|  | (K) |  |  | (K) |
| This study | −1.1 ± 0.3 | 6.6 ± 1.9 | 0.4 ± 0.1 | 1.5 ± 1.0 |
| Literature data | *−3.5 ± 3.1* | *12.8 ± 13.1* | *0.5 ± 0.3* | 2.6 ± 1.7 |
| Combined | −0.7 ± 0.1 | 3.4 ± 0.4 | 0.8 ± 0.1 | 1.9 ± 1.4 |



**Table 3.** Probabilities $\Pr(\leq \Delta T)$ deduced from Table 2 to relate a future temperature drop during an eclipse to values previously published in the literature. Bold figures are above the median, and figures in italics are below the 10% percentile of the probability distribution. The sign convention uses negative $\Delta T$ if temperature gets colder during the occulation phase.

| $\Delta T$ (K) | All | Literature | This study |
|---:|---:|---:|---:|
| –2.0 | >**0.999** | >**0.999** | >**0.999** |
| –1.5 | >**0.999** | **0.999** | >**0.999** |
| –1.0 | >**0.999** | **0.996** | >**0.999** |
| –0.5 | **0.999** | **0.985** | **0.998** |
| 0.0 | **0.967** | **0.958** | **0.961** |
| 0.5 | **0.867** | **0.908** | **0.838** |
| 1.0 | **0.719** | **0.830** | **0.644** |
| 1.5 | **0.558** | **0.729** | 0.438 |
| 2.0 | 0.412 | **0.613** | 0.268 |
| 2.5 | 0.292 | 0.494 | 0.150 |
| 3.0 | 0.200 | 0.381 | 0.078 |
| 3.5 | 0.134 | 0.283 | 0.038 |
| 4.0 | 0.087 | 0.202 | 0.018 |
| 4.5 | 0.056 | 0.139 | *0.008* |
| 5.0 | 0.035 | 0.093 | *0.003* |
| 5.5 | 0.022 | 0.060 | *0.001* |
| 6.0 | 0.014 | 0.038 | *0.001* |
| 6.5 | *0.008* | 0.023 | *<0.001* |
| 7.0 | *0.005* | 0.014 | *<0.001* |
| 7.5 | *0.003* | 0.008 | *<0.001* |
| 8.0 | *0.002* | 0.005 | *<0.001* |
| 8.5 | *0.001* | 0.003 | *<0.001* |
| 9.0 | *0.001* | 0.001 | *<0.001* |
| 9.5 | *<0.001* | 0.001 | *<0.001* |
| 10.0 | *<0.001* | *<0.001* | *<0.001* |
| 10.5 | *<0.001* | *<0.001* | *<0.001* |
| 11.0 | *<0.001* | *<0.001* | *<0.001* |