# Peer review of "Effects of vernal equinox solar eclipse on temperature and wind direction in Switzerland"

_Atmospheric Chemistry and Physics, 2017_

## Referee Comment (RC2) · Anonymous Referee #3 · 4 Aug 2017

This work provides a thorough analysis of eclipse-induced responses as observed by a Swiss network of meteorological sites. A complexity of this region is of course the range of topogaphy encountered, and the authors carefully separate the data obtained in different conditions. These reveal the wind direction and thermal changes, which are compared with previous studies. This is a valuable contribution to the literature and should be published.

In addition, the measurements in Switzerland are uniquely position to investigate the original interpretative ideas of Clayton, modified a century later by Aplin&Harrison. The scales employed are appropriate for this as they consider a large part of the European landmass away from coastal effects. Proper consideration of the remaining topographical aspects, as undertaken by the authors is therefore important in obtaining the un-

derlying effects on the dynamical structures arising from eclipse meteorology.

Minor points p1 L7 sun's disk p1 L14 comma after "After the maximum,.." p1 L27 it's –> its p9 L24 for p10 L1 mountain fig1 last line "its"

Section 2.6 Give some explanation of the consequences for the choice of the sill and range parameters

---

## Author Comment (AC2) · 14 Sep 2017

We copy in the reviewer's comments and critique in blue and provide our response in black.

*Reviewer: This work provides a thorough analysis of eclipse-induced responses as observed by a Swiss network of meteorological sites. A complexity of this region is of course the range of topogaphy encountered, and the authors carefully separate the data obtained in different conditions. These reveal the wind direction and thermal changes, which are compared with previous studies. This is a valuable contribution to the literature and should be published.*

*In addition, the measurements in Switzerland are uniquely position to investigate the*

[Figure]

*original interpretative ideas of Clayton, modified a century later by Aplin&Harrison. The scales employed are appropriate for this as they consider a large part of the European landmass away from coastal effects. Proper consideration of the remaining topographical aspects, as undertaken by the authors is therefore important in obtaining the underlying effects on the dynamical structures arising from eclipse meteorology.*

Thank you very much for this positive assessment.

*Reviewer: Minor points p1 L7 sun's disk*

This will be corrected.

*Reviewer: p1 L14 comma after "After the maximum,.."*

This will be corrected.

*Reviewer: p1 L27 it's $-->$ its*

This will be corrected.

*Reviewer: p9 L24 for*

This will be corrected.

*Reviewer: p10 L1 mountain*

This will be corrected.

*Reviewer: fig1 last line "its"*

This will be corrected.

*Reviewer: Section 2.6 Give some explanation of the consequences for the choice of the sill and range parameters*

This can be done. In fact, the sill and range parameters do not strongly affect the interpolation and the main differences between choices that we tested out were affecting the borders of the range covered with data. As an example we included the variants for all data with sill/range of $90°/10$ km, $120°/10000$ km, and $300°/1000$ km. Thus, the initial estimates for both parameters are not essential and the model fit nicely finds the best estimate. This is the ideal case if no attractors exist within the realistic domain of search of the optimization algorithm.

[Figure]

All sites

| | | |
|---|---|---|
| cyclonic | ▲ | −180°... −135° |
| | ▲ | −135°... −90° |
| | ▲ | −90°... −45° |
| | ▲ | −45°... −10° |
| | △ | −10°... |
| | ▽ | ... 10° |
| anti-cyclonic | ▽ | 10°... 45° |
| | ▽ | 45°... 90° |
| | ▼ | 95°... 135° |
| | ▼ | 135°... 180° |

**Fig. 1.** Example using a sill of 90° and a range of 10 km.

[Figure]

All sites

| | | |
|---|---|---|
| **cyclonic** | △ | −180°... −135° |
| | ▲ | −135°... −90° |
| | ▲ | −90°... −45° |
| | △ | −45°... −10° |
| | △ | −10°... |
| | ▽ | ...  10° |
| **anti- cyclonic** | ▽ | 10°... 45° |
| | ▽ | 45°... 90° |
| | ▼ | 95°... 135° |
| | ▼ | 135°... 180° |

**Fig. 2.** Example using a sill of 120° and a range of 10000 km.

**All sites**

| cyclonic | | −180°... −135° |
|---|---|---|
| | | −135°... −90° |
| | | −90°... −45° |
| | | −45°... −10° |
| | | −10°... |
| | | ... 10° |
| anti-cyclonic | | 10°... 45° |
| | | 45°... 90° |
| | | 95°... 135° |
| | | 135°... 180° |

**Fig. 3.** Example using a sill of 300° and a range of 1000 km.

---

## Author Response (AR1)

**Response to Reviewers**

We copy in the reviewer's comments and critique in blue and provide our response in **black boldface**. In gray we copied in our final author responce for reference. This allows us to provide a short response (in black) whenever we were able to do our revisions as foreseen in our final author response. We begin each response with a page and line number or range of line numbers that relate to the track changes manuscript as the reference.

During the revisions we also revised a few minor passages where we found the need for an improved wording, even when none of the reviewers was critical about it.

We hope that with these changes our manuscript can now be accepted as final paper in ACP. We thank both reviewers for the very careful and supportive assessment.

**Reviewer 1**

*Reviewer: This paper reports the effect of the solar eclipse in March 2015 on a network of measurement sites in Switzerland. The effects of topology are relevant for this region, and this is probably the most comprehensive study of eclipse meteorology over a multi-altitude network to date. The authors seem particularly interested in comparing two versions of the "cold cored cyclone" as presented by Clayton and modified by Aplin and Harrison, since the trajectory of the 2015 eclipse makes Switzerland ideal for such a test. Altogether this is a thorough and competent study, at a higher standard than many eclipse meteorology papers, and I am happy to recommend publication with some minor revisions.*

*Author Response: Thank you very much for this positive assessment.*

*Reviewer: The main concern I have is to do with the structure of the paper. The authors present their data analysis methods before describing the data analysis itself and this makes for a disjointed read. For a journal that doesn't use a "methods" section like ACP I would recommend moving the specific analysis techniques to the section on, for example, analysing temperature effects (or whatever it is).*

*Author Response: This can be done. We suggest to move the contents of Sections 2.3–2.8 in the discussion version to the respective paragraph where the results are presented (this is what this reviewer recommends in the detailed feedback below).*

**P5/L14–P6/L21: The information in these sections was moved to the respective results sections: P8/L2–8; P9/L21–25; P8/L26–33; P11/L11–12; P7/L28–32; P9/L9–14.**

*Reviewer: Occasionally the data analysis decisions do not appear to have any theoretical basis, for example, the choice of a gamma distribution for the temperature changes, and perhaps also the diurnal variation in the diffuse fraction. The gamma distribution is justified by the authors because it permits others to see their measured temperature changes in context, however this could be achieved with a cumulative probability distribution to all the data, without assuming a shape for the curves, so I am not sure what the gamma distribution really brings here. In general, the use of a purely empirical approach may not be a problem in itself, but the authors should state that this is the approach taken and explain why.*

*Author Response: Using parametric distributions in statistics, such as the Gamma distribution, has many benefits, but the reviewer is correct that there is no extremely firm theoretical basis for such a statistical approach. The Gamma distribution has a wide range of applicability, and also covers the special case of an exponential distribution. Thus, we believe that this is a good starting point for readers who do not want to use lookup tables to provide a probability estimate for a given temperature drop measured anyware to compare a new measurement with the existing ones.*

*We however see the reviewer's point and suggest to use the empirical cumulative distribution estimates in Table 3. In our discussion version these values could be calculated from Eq. (2) with the parameters given in Table 2, and thus are somewhat redundant information. With the suggested changes we would have distribution-independent information in Table 3, which is certainly an improvement.*

**P35, Table 3: we now show the empirical probability distribution and rearranged the table to have the largest temperature drops on top of the table, and the weakest (actually increase in temperature) at the bottom.**

*Reviewer: The figures are generally of good quality but occasionally the captions should be edited so that the figures can be understood without reference to the main text.*

*Author Response: We will revise the captions accordingly.*

**P21, P22, P24: These figure captions were updated to make them stand-alone without reference to the text.**

*Reviewer: The caption to figure 2 was particularly obtuse from this point of view.*

*Author Response: Originally, the individual panels were separate figures with relatively long captions. With the aggregation to one figure with five panels, we had to reduce the caption length and thus information content. Obviousely we shortened the text too much and are happy to expand it in the revision to make this figure better understandable independently from the main text.*

**P21: Caption to Figure 2 was expanded accordingly.**

*Reviewer: In Figure 8 I didn't understand why and how the probability was used – shouldn't this be explained in the main text, if it is really needed at all.*

*Author Response: The basic principle of statistical comparisons is to compare a given result (i.e. our measurements) with a potentially fully random result. As mentioned in the caption we used the uniform distribution (i.e., each wind direction change is as likely in a random system) for comparison. If our measurements do not deviate from such a random outcome, the $\Delta$Probability value is 0.00; if it is $> 0$, then our measurements indicate higher probability during solar eclipse than what we would expects in a random system (and if it is $< 0$, the probability is lower).*

*We will find a better way to describe this. We assume that using the term "probability" in the blue text items on the figure was confusing this reviewer. Of course one can always also debate on what a random outcome would be (we assumed uniform distribution), but we do not interpret this feedback in the way that this assumption was questioned.*

**P27: During the revisions we realized that using the term "Frequency" instead of "Probability" in this figure might have clarified some of the confusion. Hence we modified Figure 8 and its caption accordingly.**

*Reviewer: And on Figure 10, I (personally) think wind vectors would be a clearer way to indicate the change, which would then fit better with your figure 1. The use of colour to indicate flow directions is not intuitive.*

*Author Response: The key issue is the following: if a wind vector is presented, most readers confuse the angle of the vector with the geographic direction of wind (see example in Fig. 1 in our author response). It is almost impossible to present wind direction **differences** in the same way as absolute wind directions are presented. That's why we used symbols with colors to represent the wind direction **differences** on these panels. In general the blue–red gradient is widely used in meteorology to show negative–positive deviations from something (e.g. temperature anomalies). What we could do is to simplify the color scheme to only use the blue–red gradient with white at zero difference instead of the rainbow-color-gradient currently used in Figure 10 to be more intuitive with our color scheme.*

*Initially we of course hoped to find a way to subtract the local influence on wind direction from the measurement in a way that would leave us with the synoptic large-scale wind direction, but this proved almost impossible in the complex terrain of Switzerland; even the low-laying parts which are $\pm$flat actually experience channeled flow (as shown by Wanner and Furger, 1990, cited in our paper).*

**We showed an example in our author response. With respect to the color gradient a short investigation among colleagues suggested to keep the color gradient as is, since this is not uncommon. We suggest to keep the figure as is.**

*Reviewer: P1 L3-4 This sentence is confused between eclipse meteorology and the broader scientific benefits of studying eclipses.*

*Author Response: (actually on P2) We'll revise the text to eliminate this confusion and separate the two aspects more clearly.*

**P2/L6–7: Simply deleting this confusing sentence seems to be the best solution, hence we deleted it.**

*Reviewer: P1 L34 Should this be 1600km?*

*Author Response: (actually on P2) Yes, this was a typo, thank you for making us aware of it. Corrected.*

**P3/L8: corrected.**

*Reviewer: P2 L3-4 This sentence is ambiguous about whether a total or partial eclipse was seen at the two quoted locations. I believe the 1999 eclipse was total over south west England which would imply it was 97% at Reading and perhaps total at the other location, but please check and clarify.*

*Author Response: (actually on P3) We'll reword. See also our explanation of the confusion between partial and total eclipses below. A partial eclipse by definition is an eclipse that has no location on the Earth where totality can be seen. Thus, we need to be more clear about partial occultation during the event of a total eclipse vs. partial occultation of a partial eclipse. This will be reworded.*

**P3/L12: we now state that this was a total eclipse with maximum occultation of 97% at Reading. We also added Saros numbers to all eclipses mentioned in the text.**

*Reviewer: P4 L26 I am not sure what you mean by "model" - are you simply referring to the loess fitted values?*

*Author Response: Yes, this is the model fit. We will revise the wording to avoid potential misunderstandings. In statistics a "model" is everything fitted to the data which goes beyond the data themselves, but the term is not used in this way everywhere; e.g. "modelers" using global circulation models try to make the separation between "model" as the whole system and "modules" or "algorithms" for simpler statistical and prognostic model components. We will most likely refer to the "fit" in our revised version to avoid the confusion with the term "model".*

**P5/L6: we changed the text to read ". . . penumbral shading and these fitted values. . . "**

*Reviewer: P4 L28 is "instationarities" a proper word?*

*Author Response: The proper word is probably "non-stationarity" but we will double-check with an expert in both English and time-series statistics for the revisions.*

**P5/L8: after a careful investigation it turned out that the wording was correct, but that singular would be better suited. Hence we changed to "because of instationarity shortly before, . . . ".**

*Reviewer: P5 I recommend moving most of the material on this page to the sections where you actually discuss each effect, as explained above.*

*Author Response: This refers to Sections 2.3 to 2.8 in the discussion version of our paper. The suggested reorganisation can of course be done. We will revise our manuscript accordingly.*

**Text was reorganized as suggested, see our response with detailed page/line numbers above.**

*Reviewer: P5 L18 Can you explain what this does so that people who don't use this particular software are able to reproduce your work?*

*Author Response: Yes, we will expand this text and provide an additional general reference to bootstrapping (which can be done in many ways, but maybe the term is not yet as widely known as we thought). In short, nonparametric bootstrapping is a computer-intensive method to obtain a best estimate for statistical uncertainty (e.g. the 95% confidence interval as we do it here) by performing many simulations with subsets of data records randomly selected out of all available records. In this way uncertainty related to individual outliers or extreme values in a given dataset are becoming less important for the uncertainty estimate, and thus the uncertainty estimate obtained via nonparametric bootstrapping is a rather robust and reliable uncertainty estimate. We will explain this in more detail in the revised version.*

**P8/L30–33: This text was added to explain the basics of bootstrapping uncertainty boundaries: "Bootstrapping is an efficient computer-based method to quantify uncertainty intervals (e.g. Efron, 1979; Johns, 1988). Nonparametric bootstrapping means that the uncertainty calculations are done on randomly selected subsets of all data points available in such a way that the variation in the results obtained from many repetitions (9,999 repetitions in our case) represents the uncertainty of the estimate."**

*Reviewer: P6 L4 Why gamma? (as discussed above)*

*Author Response: See response above with our suggestion for the revisions.*

**P9/L9–14: The Gamma distribution has a wide range of applicability, and also covers the special case of an exponential distribution. Thus, we believe that this is a good starting point for readers who do not want to use lookup tables to provide a probability estimate for a given temperature drop measured anyware to compare a new measurement with the existing ones.**

**We however see the reviewer's point and thus now present the empirical cumulative distribution estimates in Table 3.**

*Reviewer: P6 L7 Both SE and SD are used for errors in this paper, can you be more consistent?*

*Author Response: The meaning of SE and SD is not the same, hence we use either or depending on what the context is: SE is the standard error of the mean and describes the uncertainty of the mean. SD is the standard deviation and describes how far a way from the mean a single observation lies. To obtain SE for the mean of a time series we would have to correct for serial autocorrelation, which is another confusion that many readers (even those with adequate statistical background) normally have, and hence we avoided to go into this aspect of serial autocorrelation by simply using the purely descriptive SD (e.g. for difference in short-wave radiation).*

*We can of course modify our text and consistently report SE also for the time series.*

**P7/L3: we now give mean $\pm$ SE for the short-wave radiation here.**

*Reviewer: P7 eq 4. Is this another example of an entirely empirical fit, or is there some reason why the diffuse fraction varies with time during the day that is not explained?*

*Author Response: Yes, this is an empirical fit. Unfortunately, the sky was not perfectly cloudless during the eclipse. That the ratio between diffuse and direct radiation is a function of solar elevation angle is well known, but we should have explained this in the text. Here we used $\Delta t$ (time difference from local noon) for simplicity, but we probably should better first calculate the solar elevation angle and then use this as the independent variable instead in order to be more physically-based. The parameter estimates would still be empirical best fits, but with elevation angle instead of $\Delta t$ as the independent estimate.*

**P7/L23–24: We tried out options to use solar elevation angle instead of $\Delta t$, but results were poorer and thus we left the empirical fit as it was, but added the text "This empirical fit was used because $\alpha$ was not a simple function of the solar elevation angle (fit not shown)."**

*Reviewer: P7 L7 Explain image analysis here rather than in the methods*

*Author Response: This will be done.*

**P7/L28–32: done.**

*Reviewer: P7 L17-20 Are you effectively working out the long wave albedo here? And if so, would it help to say that?*

*Author Response: According to Glickman, T. S. (ed.) (2000), Glossary of Meteorology, American Meteorological Society, http://glossary.ametsoc.org/wiki/Albedo the definition of albedo is: "Albedos commonly tend to be broadband ratios, usually referring either to the entire spectrum of solar radiation, or just to the visible portion." This does not include long-wave radiation and thus we do not think that a ratio between back-radiated long-wave radiation (which is a fraction of the long-wave radiation emitted by the Earth surface, not a radiation component from the sun) and emitted long-wave radiation should not be termed "albedo". We however realized that the term is used in some papers and textbook, hence we suggest to write about long-wave radiation balance and add the term "long-wave albedo" with quotes and in parentheses.*

**P8/L2: we struggled with the term "longwave albedo" but since it seems to appear in some texts, we added it in parentheses with quotes in the hope that this clarifies our text.**

*Reviewer: P7 L31 explain bootstrapping here rather than in methods section*

*Author Response: This will be done.*

**P8/L26–33: done.**

*Reviewer: P8 L4 would it help to compare the temperature changes in the literature for partial versus total eclipses, even if it is just to show there is no real difference?*

*Author Response: This comment is fully understandable and we initially also struggled with terminology. The astronomical terminology uses the terms "total", "partial" and "annular" eclipses. With all three there can be "partial occultation" (as we call it), but in the case of a partial eclipse there is no location on the earth with totality. The theoretical differences in solar short-wave radiation remaining is small between total and annular eclipses, and partial occultation of a given fraction at a site should not depend on the fact whether an eclipse is partial or total. In our understanding much of the temperature drops reported in the literature are strongly affected by cloudiness during the time of observation. This means that if we were to dwell more into analysing various factors we would have to separate the effects of total vs. annular (only very few reports) vs. partial eclipses as a function of degree of occultation and cloudiness. Information on cloudiness is however in most cases not a quantitative information that could be easily used for such an investigation and hence we decided not to add such an analysis.*

**As explained in detail in our author response, we decided not to further separate the dataset in the suggested subsets.**

*Reviewer: P8 L6-8 See comments above*

*Author Response: We will move the information from Section 2.8 here and show empirical cumulative distribution estimates in Table 3 instead.*

**P9/L9–14: done. We deleted the information about the mean and standard error of the Gamma distribution since it was not relevant for our study and may have added to some confusion before.**

*Reviewer: P8 L15 Can you take a couple of sentences to explain the normal diurnal variation in the mountain valley winds? This seems a unique local meteorology that not everyone will be familiar with.*

*Author Response: Yes, will will do this. Most likely other readers will also benefit from a short introduction on mountain valley wind systems.*

**P9/L29–33: we added the following text: "A mountain valley wind system is characterized by down-valley winds at night that contrast with up-valley winds during the day when solar irradiation on the mountain slopes leads to convective uplifting of air masses, thereby leading to up-slope and up-valley winds. At night, the radiative cooling on the same valley slopes leads to the production of cold air, which is denser than the surrounding air, and hence moves down-slope and down-valley (also known as katabatic drainage flow)."**

*Reviewer: P11 L18 Annular eclipses don't cause full occultation, and in terms of the meteorological effects are analogous to partial rather than total eclipses.*

*Author Response: We will reword. We did not claim that the effects were the same, but we wanted to express that most literature reports on temperature drops are either from total eclipses, or from annular eclipses, whereas reports on temperature drops from partial eclipses are rare. We'll carefully revise to avoid potential confusion in our statement.*

**P18/L9: we exchanged the word "full" by "maximum" to clarify: "…most temperature drop reports in the literature relate to maximum occultation during total and annular eclipses."**

**Reviewer 2**

*Reviewer: This work provides a thorough analysis of eclipse-induced responses as observed by a Swiss network of meteorological sites. A complexity of this region is of course the range of topogaphy encountered, and the authors carefully separate the data obtained in different conditions. These reveal the wind direction and thermal changes, which are compared with previous studies. This is a valuable contribution to the literature and should be published.*

*In addition, the measurements in Switzerland are uniquely position to investigate the original interpretative ideas of Clayton, modified a century later by Aplin&Harrison. The scales employed are appropriate for this as they consider a large part of the European landmass away from coastal effects. Proper consideration of the remaining topographical aspects, as undertaken by the authors is therefore important in obtaining the underlying effects on the dynamical structures arising from eclipse meteorology.*

*Author Response: Thank you very much for this positive assessment.*

*Reviewer: Minor points p1 L7 sun's disk*

*Author Response: This will be corrected.*

**We changed this throughout the manuscript (e.g.: P2/L10, 13, 14).**

*Reviewer: p1 L14 comma after "After the maximum,.."*

*Author Response: This will be corrected.*

**P2/L18: done.**

*Reviewer: p1 L27 it's − > its*

*Author Response: This will be corrected.*

**P2/L32: done.**

*Reviewer: p9 L24 for*

*Author Response: This will be corrected.*

**P11/L8: done.**

*Reviewer: p10 L1 mountain*

*Author Response: This will be corrected.*

**P12/L27: done.**

*Reviewer: fig1 last line "its"*

*Author Response: This will be corrected.*

**P20, line 4 in figure caption shows the correction.**

*Reviewer: Section 2.6 Give some explanation of the consequences for the choice of the sill and range parameters*

*Author Response: This can be done. In fact, the sill and range parameters do not strongly affect the interpolation and the main differences between choices that we tested out were affecting the borders of the range covered with data. As an example we included the variants for all data with sill/range of 90°/10 km, 120°/10000 km, and 300°/1000 km. Thus, the initial estimates for both parameters are not essential and the model fit nicely finds the best estimate. This is the ideal case if no attractors exist within the realistic domain of search of the optimization algorithm.*

**P11/L12–16: we added the following text: "We tested the range 90°–300° for the partial sill setting, and 10–10,000 km for the range setting. The results were similar due to the optimization method used in Kriging and differed only in very minor details (see examples in our response, doi:10.5194/acp-2017-321-AC2). The selection for the final computations was thus based only on the facts that a 300° (or –60°) corresponded to the typical deviation angle of wind directions under cyclonic influence, and that 1000 km covered the entire domain of Switzerland."**

**Additional Edits**

**P1, Title:** Temperature and wind direction are now in singular in the title

**P1/L17–P2/L3, Abstract:** Contents rearranged to better emphasize the main findings

**P3/L27–34:** Slightly reworded the text

**P11/L8–9:** We added a sentence on the low level jet over the Swiss Plateau (in relation to what Gray and Harrison 2016 considered a new valid interpretation of the wind direction changes observed in the UK).

[revised manuscript text omitted]